# Left powerless: A qualitative social media content analysis of the Dutch #breakthesilence campaign on negative and traumatic experiences of labour and birth

Marit S. G. van der Pijl[1]*, Martine H. Hollander[2], Tineke van der Linden[3,4,5], Rachel Verweij[3,6], Lianne Holten[1], Elselijn Kingma[7,8], Ank de Jonge[1], Corine J. M. Verhoeven[1,9,10]

1 Department of Midwifery Science, AVAG, Amsterdam Public Health research institute, Amsterdam UMC, VU medical centre, Amsterdam, The Netherlands, 2 Department of Obstetrics, Amalia Children's Hospital, Radboud University Medical Center, Nijmegen, The Netherlands, 3 Stichting Geboortebeweging (Birth Movement NL), Ede, The Netherlands, 4 GGzE, Eindhoven, The Netherlands, 5 Faculty of Behavioural and Movement Sciences, Vrije Universiteit Amsterdam, Amsterdam, The Netherlands, 6 Hechte Band, Boxtel, The Netherlands, 7 Department of Philosophy, University of Southampton, Southampton, United Kingdom, 8 Department of Industrial Engineering & Innovation Sciences, Philosophy & Ethics, Technical University Eindhoven, Eindhoven, The Netherlands, 9 Department of Obstetrics and Gynaecology, Maxima Medical Centre, Veldhoven, The Netherlands, 10 Division of Midwifery, School of Health Sciences, University of Nottingham, Nottingham, United Kingdom

* m.vanderpijl@amsterdamumc.nl

## Abstract

### Introduction

Disrespect and abuse during labour and birth are increasingly reported all over the world. In 2016, a Dutch client organization initiated an online campaign, #genoeggezwegen (#breakthesilence) which encouraged women to share negative and traumatic maternity care experiences. This study aimed (1) to determine what types of disrespect and abuse were described in #genoeggezwegen and (2) to gain a more detailed understanding of these experiences.

### Methods

A qualitative social media content analysis was carried out in two phases. (1) A deductive coding procedure was carried out to identify types of disrespect and abuse, using Bohren et al.'s existing typology of mistreatment during childbirth. (2) A separate, inductive coding procedure was performed to gain further understanding of the data.

### Results

438 #genoeggezwegen stories were included. Based on the typology of mistreatment during childbirth, it was found that situations of ineffective communication, loss of autonomy and lack of informed consent and confidentiality were most often described. The inductive analysis revealed five major themes: "lack of informed consent"; "not being taken seriously and

**Data Availability Statement:** The data used in this study has been deposited to Zenodo and can be

accessed via the following link: https://doi.org/10.5281/zenodo.3724213.

**Funding:** The author(s) received no specific funding for this work.

**Competing interests:** We declare that the organizations Hechte Band and Geboortebeweging did not provide funding to carry out this study. Two authors (TL and RV) affiliated with these organizations were part of the research team representing the client. TL is a member of Geboortebeweging, a clinical psychologist and a PhD student (GGzE & Vrije Universiteit). RV is also a member of Geboortebeweging and is a certified babywearing consultant and bonding coach (Hechte Band). Both authors were involved in managing the #GG campaign in 2016, in which they also participated themselves. TL and RV were involved in the research process from the start. They provided feedback on the research proposal and design of the study and contributed to discussing the findings within the research team and writing the paper, all from a client point of view. They were not involved in data collection or analysing the data. This does not alter our adherence to PLOS ONE policies on sharing data and materials. There are no patents, products in development or marketed products to declare.

not being listened to"; "lack of compassion"; "use of force"; and "short and long term consequences". "Left powerless" was identified as an overarching theme that occurred throughout all five main themes.

## Conclusion

This study gives insight into the negative and traumatic maternity care experiences of Dutch women participating in the #genoeggezwegen campaign. This may indicate that disrespect and abuse during labour and birth do happen in the Netherlands, although the current study gives no insight into prevalence. The findings of this study may increase awareness amongst maternity care providers and the community of the existence of disrespect and abuse in Dutch maternity care, and encourage joint effort on improving care both individually and systemically/institutionally.

## Introduction

Worldwide, women increasingly report experiences of disrespect and abuse during labour and birth, including physical abuse, non-consented care, non-confidential care, non-dignified care, discrimination, abandonment and detention [1]. Disrespectful and abusive care leads to the neglect of psychological and emotional needs in labour and birth, which are essential for labouring women [2].

Previous studies have tried to gain insight in the extent of the problem. In a worldwide study among health care workers in 2012 from the United States Agency for International Development (USAID), the most prevalent types of disrespect and abuse during labour and birth were lack of privacy, performing harmful practices, lack of information about the care, lack of informed consent and denying choice of birthing position [3]. An Italian survey among 424 women revealed that over 20% of women considered themselves victims of disrespect and abuse during labour and birth [4]. In the USA, 17.3% of the women experienced one or more types of mistreatment during pregnancy and childbirth, for example being scolded, threatened or ignored [5]. An Australian study investigating caregivers' interactions with women during labour found that women felt caregivers disregarded embodied knowledge, used lies and threats to gain compliance and prioritized their own agendas [6].

In the Netherlands, Hollander et al. asked 2192 Dutch women through social media about their traumatic labour and birth experiences; many women felt their trauma could have been reduced or prevented by more adequate communication and support by the caregiver [7]. In addition, women's choice to give birth outside professional guidelines (for example giving birth at home when a hospital birth was indicated) was often made following negative experiences with previous care [8]. These findings suggest (types of) disrespect and abuse during labour and birth are present in the Netherlands. However, scientific literature from the Netherlands on this topic is lacking.

Dutch maternity care is divided in primary midwife-led care and secondary obstetrician-led care [9,10]. In primary care, women deemed at low risk for obstetric complications are cared for by autonomous midwives antenatally and intrapartum. Autonomous midwives work independently in community practice and are assisted by trained maternity care assistants during the birth, which takes place either at home or at a birth centre. When complications occur during pregnancy, birth or postpartum, or when pharmacological pain relief is requested during birth, women are referred to obstetrician-led care in the hospital. In obstetrician-led care,

hospital-based midwives and residents provide care under obstetric supervision, assisted by obstetric nurses. Obstetricians provide care when risks or problems occur [9,11].

The Dutch "Birth Movement"(Dutch: Stichting Geboortebeweging) is an advocate for the rights of women in Dutch maternity care and is part of a global network: Human Rights in Childbirth [12]. This charity aims for "a society in which every woman is at the centre of care during her pregnancy, labour, birth and postpartum period and in which the health care system informs, facilitates and supports women in making their own informed choices" [13]. In 2016, the Birth Movement set up a campaign inviting women who had a negative experience with their care during labour and birth to publicly share their story. The campaign was initiated by a message posted on November 23rd on the social media platform Facebook, in which the Birth Movement requested women to (1) write down their story on a A4-sized paper, adding the hashtag #genoeggezwegen (known in English as: #breakthesilence or #rosesrevolution), (2) take a picture of this paper and (3) email the picture to the Birth Movement. The Birth Movement posted all pictures received between November 23rd and December 10th, 2016 on their public Facebook Page, hereby leaving out any information on the origin of the picture, such as names or email addresses. This kind of campaign originated in Spain where, known as "La Revolución de las rosas" (English: roses revolution), it was initiated in response to offensive cartoons published in the official journal of the Spanish Society of Gynaecology and Obstetrics (SEGO). These cartoons mocked women in birthing situations [14]. The campaign received global attention, inspiring similar campaigns in countries like the USA, Italy, Croatia and Germany. In the Dutch campaign, numerous stories about disrespect and abuse in maternity care were shared, which documented, amongst others, medical interventions without informed consent, intimidation, threats, violations of privacy, verbal violence and even physical violence. Many women mentioned that they had kept quiet for years and felt that this was the first time they could share their experiences. The stories evoked public and media attention, and also reached policymakers; a selection of the stories were gathered and presented, as a 'black book', to the Dutch ministry of Health [13].

The #genoeggezwegen campaign is a form of hashtag activism: the fighting for, or supporting of, a cause by using hashtags as the primary method, with the aim of raising awareness and evoking debate through social media [15]. Hashtag activism motivates citizens to share their experiences and opinions on social media, including unfiltered details and feelings [16]. Social media content enables access to information that is not easily found through scientific data collection, especially information from people who are reluctant to share such information via official ways, e.g. surveys or interviews [17]. Analysis and aggregation of social media content could therefore give valuable insight in opinions and experiences of individuals[17]. Although in general social media is increasingly being used for research purposes, studies that focus on the content of hashtag activism is relatively new [18].

Little is known about the occurrence of disrespect and abuse during labour and birth in the Netherlands. The #genoeggezwegen stories shared by women through the pictures can provide insight in negative and traumatic experiences of women with Dutch maternity care as reported by themselves. This study investigated these stories using a qualitative social media content analysis. The aim was (1) to determine what types of disrespect and abuse women reported in the #genoeggezwegen stories and (2) to gain a more detailed understanding of the experiences reported by these women in the #genoeggezwegen stories.

## Methods

### Study design

This study investigated the stories shared by women in the #genoeggezwegen (#GG) campaign using a qualitative social media content analysis.

## Data collection

In March 2019, all pictures posted by the Birth Movement on their public Facebook page were downloaded and saved on a computer by the researcher (MP). Subsequently, the textual content (#GG story) of each picture was transcribed in a word document. Some women shared their stories using multiple pictures. In this case, the textual content of all related pictures were gathered together and transcribed as one story. As this study only focused on the women's point of view, all #GG stories that were shared by caregivers, partners or other witnesses of negative or traumatic experiences were excluded. Also excluded were #GG stories that only comprised a response to the hashtag movement, without including textual content about a negative or traumatic experience. Duplicate #GG stories were identified and excluded.

## Ethical considerations

Ethics approval was sought from the medical ethics committee of Amsterdam UMC; the study was deemed not to require ethics approval (2019.402). The pictures downloaded from the Facebook page of the Birth Movement did not contain any information, such as names or email addresses of the women who shared their experiences. Although faces of individuals were visible in some of the pictures, only the textual content of the pictures was used for data analysis in this study to guarantee the privacy of the women participating in the #GG campaign. Furthermore, any identifiable information in the textual content (such as names of persons, caregivers or hospitals) were removed if present, making all data anonymous prior to analysis. All data were stored in a password-protected file.

## Data analysis

The textual transcripts of the #GG stories were copied into MAXQDA Analytics Pro, a qualitative data analysis software program [19], in order to manually assign codes and categories to (parts of) the textual content. First, the characteristics of the stories were investigated and described (e.g. number of stories, length of stories, type of content, type of interventions mentioned, type of caregivers mentioned). Then, the qualitative content analysis was conducted in two ways: (1) The textual content was deductively coded using the existing typology of mistreatment of Bohren et al. [22]. A deductive coding process is based on previous knowledge and/or theories and can be used to test the applicability of an earlier theory in different situations [20,21]. The typology of Bohren et al., based on analysis of 64 studies from 34 countries, consists of seven types of mistreatment of women during childbirth (Table 1) [22]. The types and subtypes of mistreatment as described by Bohren et al. were used as codes to categorize the textual content of the #GG stories, thereby determining what type of mistreatment was reported therein, taking the women's own words at face value. When more than one type of mistreatment would fit one (section of a) story, multiple codes were used.

(2) Next, an inductive coding procedure was used. The primary purpose of an inductive coding procedure is to allow research findings to emerge from the raw data without restrictions [21].This procedure was used to disentangle, examine, compare and categorize the stories into codes independently from the typology of mistreatment of Bohren et al, allowing new themes to emerge from the data. The first 50 stories were coded and categorized by two researchers together (MP and MH). Any discrepancies in coding were discussed and consensus was found. The remaining stories were analysed by one researcher (MP). After the coding of the first 50 stories, (sub)categories were created by comparing codes and merging codes together. After analysing every 100 stories thereafter, the formulated (sub)categories were adjusted if needed (MP). In the last phase of coding, the codes were structured; five main themes were established; and a coding tree was created (MP). The themes that emerged from

**Table 1. Typology of mistreatment of women during childbirth of Bohren et al.**

| Types of mistreatment | | |
|---|---|---|
| Physical abuse | Use of force | Using rough touch or extreme force; aggression, punching, hitting, slapping. |
| | Physical restraint | Women restrained to the bed or gagged during delivery |
| Sexual abuse | Sexual abuse | Sexual abuse or rape. |
| Verbal abuse | Harsh language | Harsh and rude language; judgmental and accusatory comments. |
| | Threats and blaming | Threats of poor outcome of child; threats of withholding treatment; blaming women for child's poor health outcomes. |
| Stigma and discrimination | Discrimination based on sociodemographic characteristics | Discrimination based on ethnicity, race, age, education level, social class and/or income level. |
| | Discrimination based on medical conditions | Discrimination based on women's existing medical conditions, for example HIV. |
| Failure to meet professional standards of care | Lack of informed consent and confidentiality | No consent asked prior to a procedure; not adequately informing of the risk and benefits of a procedure; breaches of confidentiality. |
| | Physical examinations and procedures | Painful vaginal exams (possibly in non-private settings, without consent or communication); refusal to provide pain relief. |
| | Neglect and abandonment | Leaving women alone, ignored and abandoned; long delays in care; no skilled attendant present at time of delivery. |
| Poor rapport between women and providers | Ineffective communication | Poor communication; poor staff attitudes; being 'left in the dark'; dismissal of concerns; language and interpretation issues. |
| | Lack of supportive care | Not adequately cared for; lack of comfort and empathy; denial of birth companions. |
| | Loss of autonomy | Women treated as passive participants during childbirth; forcing women to deliver in undesirable or humiliating positions; denial of safe traditional practices; denial of food, fluids or mobility; detainment in facilities. |
| Health system conditions and constraints | Lack of resources | Dirty, noisy, overcrowded facilities; staff shortages; unskilled staff; inadequate medical supplies; lack of privacy. |
| | Lack of policies | No (safe) opportunity for expressing opinions about the treatment and services; lack of accountability and sanctioning within the health system. |
| | Facility culture | Bribery and extortion of women; unclear fee structures; unreasonable requests of women by care givers |

the data were discussed within the research team, including two client representatives, and adjusted thereafter. Data saturation was reached after analysing 350 stories, which was confirmed by the analysis of the remaining stories.

The results describe how women reported their experience in the stories. Hereby we take the women's words at face value. That means, for example, that if the woman described in the story that she was left alone or that there was not enough staff, that we report this in those terms—and equally if she describes her feelings, for example that she felt lonely or alone, we describe those feelings as such. The quotes presented in the findings were translated from Dutch to English by two of the authors (MP and EK).

During the data analysis process, it was not possible to validate the interpretations of the data by performing a member check, due to the anonymity of the women who shared the #GG stories. However, the analysis was checked by the two client representatives in the research team, both of whom participated in the #GG campaign and are members of Birth Movement.

## Research team and reflexivity

It is important to reflect on the social positioning of the researchers involved and the way in which their beliefs may have informed the research. Two female researchers took the lead in the data analysis process (MP and MH). MP is a PhD researcher in the field of Respectful Maternity Care with a background in health sciences and global health. She has no medical background and no experience of pregnancy and birth herself. MH has a medical background

in both midwifery and obstetrics; a PhD in traumatic birth experiences/birthing outside the guidelines; and has positively experienced labour, birth and motherhood herself. Both researchers (MP and MH) had previous knowledge of research in the field of public health and negative and traumatic birth experiences, and had experience in analysing qualitative data prior to this study. Furthermore, both researchers have specific interests in women's autonomy and rights, both in general and in maternity care. Both researchers were aware that their roles impacted the way they interpreted the data. Therefore, the process of the data analysis was discussed within the research team, leading to greater trustworthiness of the interpretation of the data. The other members of the team included three midwife researchers (CV, AJ and LH), two client representatives (RV &TL) and a philosopher/ethicist with a degree in medicine and psychology (EK). All members were female and all but one experienced labour, birth and motherhood themselves, with experiences ranging from very positive to very negative.

## Results

In total 533 stories were collected, of which 95 were excluded, leaving 438 Stories for the analysis. The excluded stories consisted of duplicate stories (41), stories only expressing gratitude to the movement (12) and stories that were written from partners' (14) or caregivers' perspectives (28) (Fig 1). The 438 included stories varied in form (e.g. poetry; quotations; or bullet-point description); use of language; and length. The shortest story comprised three words and the longest 311. In almost all stories, women described a situation during childbirth or the postpartum period, with five exceptions. One described contact with health care professionals during pregnancy, one an encounter at the pharmacy during pregnancy, one IVF treatment, and two filing a complaint concerning previously received care. Most stories described multiple events including different types of situations; some only shared one or two sentences about one particular situation. A type of caregiver was mentioned 334 times in the stories, with some women mentioning multiple types of caregivers within one story. Obstetricians (97 times) and midwives (either hospital-based or community, 90 times) were mentioned most often, followed by nurses (68 times). In 314 stories, at least one type of intervention was mentioned by women. Episiotomy (63 times), caesarean section (51 times), suturing (48 times) and vaginal exams (44 times) were mentioned most often.

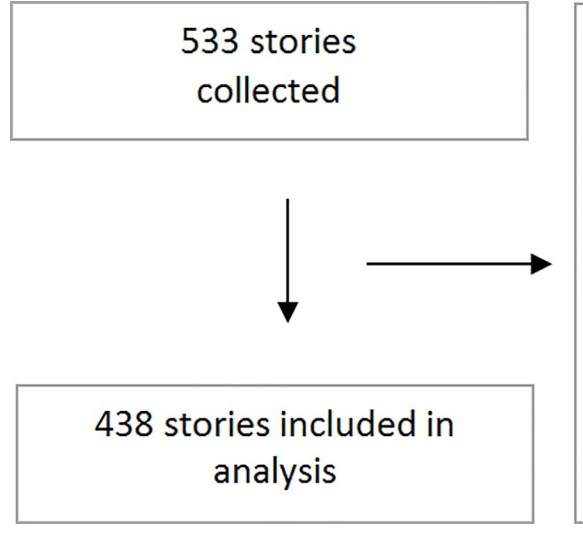

**Fig 1. Flowchart of the #genoeggezwegen stories included for analysis (n = 438).**

## Deductive coding procedure by the typology of Bohren et al

Table 2 shows what types of mistreatment according to Bohren et al.'s typology were coded in the stories. Since in most stories women described multiple situations and/or types of mistreatment, most stories were given multiple codes. Ineffective communication (coded 201 times) was found most often, followed by loss of autonomy (coded 146 times) and lack of informed consent and confidentiality (coded 139 times). We only coded sexual abuse when the stories explicitly mentioned it as such. Other situations that may also have warranted coding as sexual abuse, such as internal exams against/without women's permission (e.g. when the exam was unannounced and roughly performed and/or women's refusal or request to stop was ignored); internal exams that were perceived as unnecessary; or painful exams without compassion, were coded as physical abuse in keeping with taking the women's words at face value, even though in some of these cases the coders got the strong impression that women implicitly conveyed an experience of sexual abuse. Discrimination based on gender was not included in 'stigma and discrimination' because childbirth only happens to women and therefore it was difficult to disentangle whether the experience occurred because of gender inequality.

## Inductive coding procedure

Five major themes emerged from the stories: "Lack of informed consent"; "Not being taken seriously and not being listened to"; "lack of compassion"; "the use of force"; and "short and long term consequences". The code tree is shown in Fig 2. We identified one overarching theme that connected all major themes: "Left powerless".

**Table 2. The #GG stories coded according the typology of mistreatment of women during childbirth of Bohren et al.**

| Typology of mistreatment | | Times of coding |
|---|---|---|
| Physical abuse | | **50** |
| | Use of force | 41 |
| | Physical restraint | 9 |
| Sexual abuse | | **4** |
| Verbal abuse | | **70** |
| | Harsh language | 41 |
| | Threats and blaming | 29 |
| Stigma and discrimination | | **7** |
| | Discrimination based on sociodemographic characteristics | 7 |
| | Discrimination based on medical conditions | 0 |
| Failure to meet professional standards of care | | **318** |
| | Lack of informed consent and confidentiality | 139 |
| | Physical examinations and procedures | 95 |
| | Neglect and abandonment | 84 |
| Poor rapport between women and providers | | **427** |
| | Ineffective communication | 201 |
| | Lack of supportive care | 80 |
| | Loss of autonomy | 146 |
| Health system conditions and constraints | | **63** |
| | Lack of resources | 22 |
| | Lack of policies | 4 |
| | Facility culture | 37 |

**Lack of informed consent**

- No choice in care given / caregivers follow standard protocol of care
- Informing of, but not asking women's consent prior to, intervention
- No communication at all prior to intervention
- Ignoring women's refusal or request to stop during intervention
    - Women lacking or losing control in labour and/or birth
    - Overwhelmed / distraught / surrendered / defeated
    - Insecure / felt as failing / ashamed

**Not being taken seriously and not being listened to**

- Ignoring women / not providing information about birth progress / denying women's requests
    - Fear / Feeling abandoned
- Denying pain relief when requested / no (working) anesthesia given in operating theatre
- Separation of mother and baby after birth / mother having no say over the baby
- Dehumanization and objectification

**Lack of compassion**

- Belittled
- No introductions by caregiver / no compassion / rude / being ridiculed
- Threatening with intervention / threatening with bad outcomes baby
- Demanding something from woman
- No privacy given / women being watched
    - Humiliation
    - Upset with caregiver

**Use of force**

- Pushing on the abdomen and/or 'violently' pulling out the baby / placenta
    - Feeling horrified / thinking of dying / torture
- Using excessive force during intervention
    - Feeling violated

**Short and long term consequences**

- Trying to make sense of what happened / wondering why
- Consequences and thoughts
- Post-Traumatic Stress Syndrome / Postpartum Depression / needed treatment
- Scared to have more children
- Reliving experience

**Fig 2. Coding tree sorted by the five main themes emerging from the inductive analysis of the #genoeggezwegen stories.**

## Lack of informed consent

In more than a quarter of stories, women reported at least one act performed by caregivers without informed consent. Some described a general lack of choice in care, for example due to caregivers' following hospital protocol.

> *[. . .] I was hooked onto the IV. I was opposed to this, but it was hospital protocol. The nurse said casually that if I were to refuse, perhaps I could try for a different hospital? [. . .] (story 143)*

Sometimes situations were reported in which caregivers only announced their intention to perform an intervention, but neither consulted nor asked consent of the woman.

> *During my super-smooth second birth, the resident suddenly appeared, and said: "I will do an epi during the next contraction". "Consultation? Nought. Zero consultation took place. [. . .] (story 63)*

In other situations, women described caregivers carrying out procedures such as internal exams, episiotomies, or amniotomy absent any prior communication with the woman.

> *Without consultation, consent, or even letting me know, they cut me open. [. . .] (Story 213)*

> *I am lying on my back with my legs up in stirrups. Without saying a word, the midwife shoves her hand inside me. I recoil in pain. Dumbfounded, I cannot utter a word. When the midwife notices my angry face, she says: "Oh, yeah, I had to stretch you out for the expulsion". (story 511)*

In approximately one third of all situations in which actions without consent were described, women reported having said 'no' explicitly either prior to the intervention, or asking the caregiver to stop during the intervention. Despite this, the intervention was still carried out or continued.

> *Even though I had emphatically prohibited the doctor from breaking my waters without permission, she did it anyway. [. . .] (story 339)*

> *My birth plan said: Stop the vaginal examination when I say 'stop'. A vaginal examination is being performed. I say: "Stop". She does not stop. My husband says: "Hey!—she says stop!" She just carries on. (story 70)*

Women reported that acts performed without informed consent resulted in feelings of loss of control including: being overwhelmed, feeling distraught or feeling defeated. A loss of autonomy and confidence in the process was also described.

> *[. . .] I felt helpless and defeated. I lost all control and, with that, my self-esteem. (story 261)*

## Not being taken seriously and not being listened to

The second theme appeared in almost half the stories and includes stories in which women report being ignored and not being taken seriously, and stories about women who experienced unsupportive care and/or being left alone. Women described caregivers' leaving them alone during labour and birth or postpartum, most commonly in the immediate postpartum period, in which a delay in care was reported. For example: being left alone in the labour room after

birth, or in the recovery room after a caesarean section, without knowing where their partner or baby was and/or how their baby was doing. Other women reported being left alone all night after birth, without any help for them or the baby.

> [. . .] After giving birth I was left alone with my feet up in stirrups. After two hours somebody said: "Oh, you are still here". [. . .] (story 295)

Many women described that being left alone by caregivers made them feel abandoned and scared. Postpartum, women felt no longer important now that the baby was born.

> q[. . .] Not knowing if my baby (who had a brain defect) was okay or going to make it. 50 mins I lay there [in the recovery room]–alone and forsaken–in fear. (story 57)

> [..] I felt discarded, like a disposable wrapper. (story 384)

Women also reported caregivers' not talking *to* but *about* them, caregivers' not taking notice of their presence while being in the room, caregivers' talking 'in between their legs' to each other, and caregivers' not paying attention to the woman as the person giving birth.

> The placenta didn't want to come out. Rushed to the operating theatre; lost liters of blood. [. . .] Suddenly somebody whips the blankets of my naked body, pushes my legs aside. Next I hear panicked deliberation: "if the bleeding continues like this, she might die". Hello! I can hear you!! (Story 272)

> Literally in between my legs, they are deliberating about an episiotomy. Nobody asks me anything. (story 332)

Women also reported caregivers talking about non-relevant topics during their labour and birth, even during situations they perceived as an emergency.

> On the way to the operating theatre for an Christmas eve caesarean section. My baby was in danger. The staff were talking in detail about their presents and plans. (story 451)

Some women described trying to talk to their caregivers about their sensations in labour, but caregivers did not listen or take them seriously. Contexts described by women frequently involved feeling a lot of pain at the start of induced labour or feeling the urge to push. Caregivers' ignoring of women's worries about the progress of labour was also reported.

> My waters broke and straight away I had MASSIVE contractions. "This is nothing yet–it could easily take another week" said the OB. The pain made me vomit. The hospital midwife refused to check my dilation. I asked more than 10 times, and when she finally relented: 9 centimeters. 10 minutes later, my daughter is born. (story 248)

Another form of caregivers' ignoring of women was the dismissal of requests. Requests were often related to caregivers' postponement of, or not doing, certain interventions, or were related to caregivers' care in the immediate postpartum period. A specific request often reported as being ignored was the request for pain relief, both during labour and postpartum.

> [. . .] Unbearable pain; I was panicking. Every request for an epidural was ignored. "Your body will never give you more pain than you can handle". [. . .] (story 275)

*[After a long and exhausting labour]: 14 Stitches. "we can manage that on the numbing agent given for the episiotomy". I have never experienced that much pain in my life! She said that was impossible–and continued suturing. (story 351)*

In the case of an emergency caesarean section, some women described feeling that the epidural did not work properly. They reported that they expressed this to the caregiver, but were ignored.

*I was transported to the operating theatre for a caesarean. I keep telling them that the epidural is not working, and I keep puffing away my contractions. Nobody responds. When they pinch me to check [the epidural], I report a shooting pain, and again that the epidural is not working. After a few more attempts the doctor's patience has run out, and says: "you feel something, but no pain". Then, despite my vehement protesting, he starts the operation. When I howl in excruciating pain, and, in tears, tell them they have to stop, the doctor says: "you are not in pain, you are just afraid". (story 375)*

Another request often reported as disregarded by caregivers was the request to see, hold, or feed the baby after birth. The context often involved a caesarean section or admission of mother and baby to separate wards.

I had an emergency caesarean. *My healthy son was born. I was briefly allowed to see him, then he was taken to neonatology. He had a slight fever. After 13 hours, and insisting, I was allowed to hold him. They could have given me any baby. (story 288)*

Women described that caregivers' ignored them, did not take them seriously, and caregivers' dismissals of their requests made them feel dehumanized and objectified: e.g. 'feeling like a number' among many others giving birth, and 'feeling like an animal or an object' during childbirth.

*[..] There were about nine people standing around me but I never felt so alone in my life. (story 481)*

*[..] I felt neither seen nor heard. I, as laboring woman, did not matter. [. . .] (story 260)*

## Lack of compassion

In more than half of the stories, women reported caregivers' lack of compassion. Most often: caregivers telling women to 'stop acting like a baby' or to 'grit your teeth and bear it', for example when they indicated pain or exhaustion during labour and birth.

*Lying on my back there was nothing I could to cope with the suddenly intense contractions—I wanted an epidural. "Madam, please stop whining". (story 112)*

Sometimes women reported that caregivers neither introduced themselves, nor communicated with the woman directly, upon entering the labour room. Other women experienced caregivers' contact with them as lacking in compassion, or that caregivers talked to them in a rude manner. During birth, women described being told to be quiet when they made noises; to push in a certain way; to work harder during the pushing phase, and being screamed at.

*I am coping with the contractions in my own way when a 'white coat' walks in [. . .] who says: "you really don't need to make so much noise". Suddenly I am very conscious of my*

*surroundings, and focused more on what others think of me, than on myself. My contractions fade away and in the end it is a caesarean. (story 133)*

Women also reported caregivers' using threats. For example: caregivers threatened forced interventions if women wouldn't listen or cooperate, or poor health outcomes for the baby if women expressed their own preferences or refused an intervention.

*"I don't understand that you don't want this. You don't want the baby to die in your belly, do you?" After I refused a scalp electrode. (story 419)*

Other women described caregivers having certain expectations during or after birth counter to the women's preferences. During birth this is often related to the way of pushing or the birthing position. Women reported about caregivers telling them to shower on their own after giving birth, despite these women felt they were not up to the task.

*The midwife made me to lie flat on my back with my legs in stirrups. I hate this position. (story 391).*

*I had to shower right away. Eating first wasn't possible. I fainted in the shower. Nurse sighed: "Now you'll have to stay the night too." [. . .] (story 295)*

Descriptions of insufficient compassion by caregivers included caregivers' downplaying of a woman's traumatic experience or bad outcome after the birth.

*[A horrifying labour due to medical error] "Oh well, at least you've got two [children]"–so said the obstetrician. I was never allowed to be pregnant again. [. . .] (story 294)*

Women reported that caregivers' negative attitude and poor use of language left them feeling unsupported, humiliated and belittled.

*I needed a brief moment of compassion, support and encouragement when I was almost pushing [..] I got a threat from the obstetrician instead: "Ms, I am preparing the vacuum pump. You have to hurry." (story 494)*

*At 08.15 I was being sutured. Every prick in my battered vagina, I felt. When I asked for pain relief, she said: "I could also leave it as it is, if you prefer." I felt so small! (story 239).*

Other women felt angry due to caregivers' attitude, most often during the active phase of labour.

*[..] The only strength I could find was rage, and in rage I gave birth to my baby. [. . .] (story 282)*

## Use of force

In 1/5th of stories, women described the use of force, mostly during the active stage of labour. Descriptions included caregivers' use of force prior to or during interventions. For example: an intervention being carried out anyway, despite the woman's refusal and/or attempts to prevent it from happening by e.g. closing her legs, kicking and/or screaming.

*I was involuntarily catheterized during birth. I yelled NO. According to the nurse, this is 'how it is done'. Two people were holding down my legs. (story 103)*

Some women experienced these types of situations as sexual assault, for example when caregivers overruled women's refusals and/or carried out forced internal exams without compassion.

*[. . .] While I was laying there, in all my naked vulnerability, she barked at me, twice: I should not interfere. With her hand in between my legs, looking me straight in the eyes, she said that. It felt like violation. [. . .] (story 517)*

Women also described caregivers using force to get the baby or placenta out, for example by pushing on the abdomen or pulling on the baby or the umbilical cord.

*After 15 minutes the placenta is still not here. The resident is leaning on his elbow, pushing his full bodyweight into my abdomen. I am crying. Then he pulls on the umbilical cord. He thinks he needs so much force that he has to put one foot on the bed for leverage. (story 347)*

Women reported that the use of force made them feel horrified, variously described as going through 'horrible circumstances', 'gruesome situations' and it feeling like 'torture'. Some reported feeling like they were dying.

*She used full force on my abdomen. . . she was pushing so heavily. . . I experienced so much pain. . . I closed my eyes and thought: I am not going to survive. (story 35)*

## Short and long term consequences

In almost a quarter of all stories, women described short- and long-term consequences of the experience. Emotional trauma was the most frequently reported consequence of these experiences, sometimes characterized as unnecessary and deeply rooted. Some women described they still suffer the emotional consequences of the event years later.

*[. . .] More than three years later, panic still overwhelms me when I think about this. Every single time, with full force. (story 38)*

*[. . .] I don't know what my baby felt, smelled or sounded like; the sound of the episiotomy scissors I remember in detail, 2.5 years later. (story 319)*

A few women are still dealing with the consequences of interventions that took place, of which internal exams were most frequently mentioned. Women reported difficulty sleeping; experiencing nightmares; or feeling sad when they think about their experience.

*[. . .] It felt like rape, I can still feel the hand (inside me). (story 237)*

*[. . .] The suturing was very traumatic and painful; it couldn't sleep for months, thinking about it. (story 356)*

Some women specifically described post-traumatic stress disorder (PTSD), postpartum depression or the need for psychological treatment in their stories.

*[. . .]This was the beginning of my postnatal depression. (story 386)*

While some women tried to make sense of what happened to them in the stories (e.g. by giving reasons such as unavailability of caregivers, or hospital protocol), others express wonder about why the situation occurred.

*After the induction—which I did not want–had started, we rolled from one intervention into the next, everything just got worse and worse. Why did nobody listen to me? At no point was I or my son in danger. Then why all these forced interventions? (story 362)*

Some women reported that their experience left them scared to get pregnant or give birth again.

*[. . .] I still dream about it. I am too scared to try for a second (child). (story 197)*

## Overarching theme: Left powerless

Despite the variety of women's descriptions of their negative and traumatic experiences, a feeling of powerlessness occurred in all five main themes and was therefore identified as the overarching theme.

*[. . .] Never in my life have I felt so powerless. [. . .] (story 109, story 208, story 407)*

When informed consent was lacking, women reported a feeling like their power was taken away from them by caregivers as interventions took place without their knowledge and/or permission. In cases of being ignored or left alone by caregivers, women described their own counteractive efforts at keeping or regaining control, which were not heard or dismissed–leaving them powerless. Lack of compassion was reported by women, leaving them feeling belittled and humiliated and where caregivers used force, women described being horrified by the acts that were performed, and feeling completely powerless.

Women not only described being left powerless during the situations themselves, but also afterwards: the emotional trauma ascribed to their labour and birth experience continued to haunt them; women stated they will never forget their experience.

*[..] I lost all control, all emotional feeling. The baby is just a child, not mine. The black pit that I fell into lasted several years. Now, eight years later, occasionally it still gets me. (story 170)*

## Discussion

The present study examined women's negative and traumatic experiences of Dutch maternity care, as described in the #genoeggezwegen stories. First, we carried out a deductive coding procedure using the typology of Bohren et al. to determine what types of mistreatment were mentioned in the #GG stories. We found that ineffective communication, loss of autonomy and lack of informed consent and confidentiality were mentioned most often by women. An additional separate inductive coding process allowed us to further investigate the content of the stories, and revealed five main themes: lack of informed consent; not being taken seriously and not being listened to; lack of compassion; the use of force; and short and long term consequences. Feelings of losing control, fear, being objectified and being humiliated were frequently reported as experienced by women. 'Left powerless' was identified as overarching theme as it occurred in all five main themes.

Most current literature on disrespect and abuse during labour and birth focuses on facilities or hospitals where obstetricians are the responsible caregivers [22–25]. In the #GG stories, midwives and obstetricians were mentioned by women in roughly equal numbers. This may indicate women experience disrespect and abuse throughout the Dutch maternity care system,

regardless of type of caregiver. Thus, awareness-raising efforts must be targeted at all caregivers in the system. Furthermore, in the majority of stories, women described one or multiple types of interventions. Most often it was not the intervention itself that was identified by women as the direct cause of her negative and/or traumatic experience, but rather the context in which the intervention took place. This is in line with previous studies stating that caregivers' interactions are more important for a woman's birth experience than the interventions she undergoes [7,26,27]. Even so, the types of interventions most frequently mentioned in these stories may indicate what can be especially vulnerable moments during labour and birth according to women, in need of extra attention when striving to provide respectful care.

For the deductive coding process, the typology of mistreatment by Bohren et al. was found to be helpful in identifying what types of mistreatment were reported. Often, one story fitted multiple types of the typology of mistreatment, underlining the complexity of the occurrence of disrespect and abuse during labour and birth [1,28]. According to Bohren et al., acts of mistreatment in maternity care can be active or passive, related to the behaviour of individual caregivers or related to health system conditions [22]. The variety of situations described in the #GG stories confirms this.

The inductive coding analysis of the #GG stories allowed certain themes to emerge from the data, enabling a broader understanding of the experiences described, including women's feelings and their consequences. Lack of informed consent was an important theme, consistent with other studies [29–31], and came in multiple forms: women described not being informed or consulted prior to procedures; not being asked for, or given time to, consent; or having their explicit refusal overridden. An Australian study similarly found that women reported a lack of informed consent prior to multiple types of procedures in maternity care [32]. Since 1995, informed consent for medical procedures is a legal requirement in the Netherlands. The act states that the patient needs to be informed about the content, the goal, the consequences and alternatives of the procedure and consent always needs to be obtained; explicitly for invasive and surgical procedures. This makes the experienced lack of informed consent, as described by women in the stories, particularly striking. Not only is this an aspect of respectful maternity care; it also a legal requirement under Dutch law, suggesting some interactions in Dutch maternity care may violate the law [33]. Previous literature from Australia found that maternity caregivers have a poor understanding of their legal accountability and women's rights, causing them to inconsistently support women's rights during pregnancy and birth [34].

Women described in the stories that a lack of informed consent often caused feelings of loss of control and loss of autonomy. In a survey on Dutch women's traumatic labour and birth experiences, women most often attributed their traumatic experience to the lack and/or loss of control during labour and birth [7]. In the last decade, there has been an increase in awareness of the autonomy and wishes of patients in health care provision. Where previously caregivers played a dominant role in decision-making about patients' health care, today patients are taking a larger role [35]. This shift is visible in a systematic review on decision making in general health care, in which 70% of the patients preferred shared decision making roles in studies after 2000, compared to 50% of the studies before 2000 [36]. Active patient engagement has been shown to be beneficial for patients' health outcomes, satisfaction of care and the feeling of autonomy [37]. The importance of autonomy and control is emphasized by a qualitative analysis of Dutch birth stories on the social media platform Instagram [38]. The main theme identified here was 'doing it yourself'. This captures the notion that women's aims in sharing their birth story online was not just to represent their birth, but also to explain to what degree they were in charge of their birth; strong emotions were associated with doing and deciding

things for themselves [38]. A sense of choice and control during birth therefore appears a crucial element in achieving a positive experience.

A second main theme in the #GG stories was women describing not being listened to and not being taken seriously by caregivers, which resulted in women feeling sidelined and objectified in their own care. A lack of compassion of care providers experienced by women was another important theme, leading to feelings of fear, humiliation and anger. This confirms previous studies that emphasise the considerable effect of patient-caregiver interaction on women's experiences [6,26,30,39]. Caregivers' responding to women's needs and offering emotional support to women have been found to promote positive birth experiences [40]. In the theme use of force, violent care was reported by some women, triggering major emotional consequences. This is in line with previous research, showing that women's accounts of birth trauma often included violence and physical abuse [6].

Caregivers may not be aware of the major influence they have on women's experience; they sometimes consider their actions routine and fail to realise that women can experience them as traumatic [6]. Reed, Sharman and Inglis reported that dehumanizing practices in maternity care can be so pervasive, that care providers cease to be able to perceive them as such [6]. Some caregivers explain their unawareness in terms of a disproportionate focus on the biomedical care in educational institutions and health care facilities, at the expense of the humanistic aspects of labour and birth [31]. However, even caregivers that realise the potential impact of their actions experience barriers to providing respectful maternity care due to their work conditions (e.g. long hours, no breaks), their responsibilities and duties (e.g. heavy workloads, giving care to multiple patients at a time), and/or local work culture (e.g. experienced hierarchy, strict protocols). All of these can lead to stress and sometimes even traumatic experiences for themselves [41,42]. Women occasionally report the presence of such circumstances in the current study: women mentioned busy caregivers that have to care for multiple women at once; caregivers that appeared rushed; and strict adherence to protocols. Healy Humphreys and Kennedy found that strict protocols and guidelines often serve to undermine women's autonomy, which can evoke feelings of conflict among caregivers. Caregivers state that they are afraid of being criticised by other health care professionals; good outcomes are never celebrated, while bad outcomes are put under a spotlight. This prevents caregivers from providing care that promotes a positive birth experience [43].

There are therefore many factors that stand in the way of providing respectful care. It is important that, in addition to considering the level of individual interactions between women and caregivers, we consider the structural challenges present in facilities and health care systems [24,44]. In addition, the underlying problem may not be specific to health care, but rooted within wider culture. Disrespect and abuse during labour and birth can be seen as a form of sex and gender discrimination and/or violence, as it specifically affects the health and rights of women [45,46]. This emphasizes that we also need to look at the broader aspects of disrespect and abuse during labour and birth, emphasizing the responsibility as a society to tackle gender inequality and promote respectful maternity care [24,44,47].

'Left powerless' was identified as the overarching theme representing women's experiences in the current study, concurrent with previous studies. Elmir, Schmied and Wilke described that feelings of powerlessness are caused by many different situations during labour and birth, e.g. women having no say in what happens during birth; not being provided with information; or not being able to make informed decisions [48]. Other research shows that feelings of powerlessness are associated with the development of psychological trauma and (postpartum) PTSD [39,49]. Stramrood et al. conducted a survey study among 907 Dutch women and found that 1.2% of respondents suffered from PTSD following labour and birth, with 9.1% experiencing one or more PTSD symptoms [50]. In the #GG stories, some women also mentioned post-

traumatic stress symptoms and other severe short and long term consequences of negative and/or traumatic experiences with maternity care, which was identified as the fifth theme of the study. The major and long-term consequences of women's negative and/or traumatic experiences reported in the #GG stories indicate the need for increased attention to women's psychological wellbeing during labour and birth. The most recent WHO intrapartum guidelines make a similar recommendation, and recognize a positive childbirth experience as a main end point for all women undergoing labour [2].

#GG is not the only hashtag campaign on social media that has covered women's issues; others include #metoo, #notokay (sexual harassment), #whyIstayed (domestic violence) and #shoutyourabortion (abortion) [51–54]. Bogen et al. investigated the use of the #metoo hashtag on Twitter and concluded that this platform facilitates a space where individuals can share personal trauma and connect with others with similar experiences. They pointed out that sharing on Twitter can raise awareness of sexual violence [53]. The present analysis confirms that social media content can be usefully investigated in order to gain a better understanding of women's experiences.

## Strengths and limitations

To our knowledge, this is the first study that analysed the content of one of the worldwide #breakthesilence or #rosesrevolution campaigns. It shows that content shared as a form of hashtag activism can provide important information on experiences, including unfiltered details and feelings.

Both a deductive and an inductive open coding procedure was used to analyse the stories. The deductive approach showed the applicability of using existing categories to code the #GG stories, whereas the inductive approach led to a more detailed understanding of the negative and traumatic experiences shared in the #GG stories. Using both approaches allowed us to not only identify the types of disrespect and abuse that were experienced by women, but also to describe feelings and consequences related to women's experiences.

For the present study, all #GG stories that were posted on the public Facebook page of the Birth Movement were analysed. However, the huge amount of social media engagement around the #GG campaign also motivated some women to share their experience directly on social media, for example in the comment section of Facebook messages, through personal Facebook pages or on other social media platforms such as Twitter or Instagram. The amount of stories analysed in this study could therefore underrepresent the number of actual #GG stories, by missing stories that women shared in other ways.

In the #GG campaign, women had limited space available to write their story as they were asked to share it on one A4-sized paper. Due to this, the stories were short and often decontextualized. It is also important to note that the present analysis took the women's stories at face value, which means it can only represent the woman's perspective of her negative/traumatic experience with maternity care and its context and causes. It is quite possible that the caregivers involved or bystanders would have had a different perspective on some of these situations and their causes. It is also likely that actions and interactions were often not intended as they were received, nor do we know the amount of time elapsed between the woman´s experience and her writing of the story, which may have led to recall bias. If some stories are quite old, then they may stem from an earlier era that no longer represents present-day maternity care. Whilst these considerations should make us wary of jumping to conclusions about causes or caregivers' knowledge, intention, or insight, the #GG stories provide us with useful insight into the negative and traumatic experiences from the women's point of view. Furthermore, social media users are not representative of the entire population, and the sample included

here was heavily self-selected. The #GG campaign was started by the Dutch Birth Movement, whose goal is defending women's rights in maternity care. It is likely that a large group of participants of the #GG campaign had a specific interest in subjects such as women's rights, and were therefore motivated to participate in the campaign. There are no demographic data about the participants of the #GG campaign, therefore the findings of the study cannot be generalised to the general Dutch maternity population. Even so, the present study provides detailed and valuable insight in the experiences of Dutch women who did participate in the #GG campaign. This gives reason to think that disrespect and abuse do occur during labour and birth in the Netherlands, and can form the basis for further investigation.

## Recommendations

Women's control during labour and birth; being seen and heard; and being provided with support are central elements of good maternity care. These aspects, as well as a better understanding of the importance of informed consent, should be explicitly included in the curriculum by all educational institutions relevant to maternity care provision. A positive labour and birth experience should be recognized in both the health care system and by the community as a main endpoint for maternity care, in line with WHO recommendations. Only then can we create an environment in which achieving such an experience is a priority amongst others, which is one aspect of reducing disrespect and abuse in maternity care.

We also recommend (1) conducting a quantitative study among a group of women representative for the general Dutch population, to obtain information on the incidence of disrespectful and abusive care in the Netherlands and (2) a qualitative study on causes of the occurrence of disrespect and abuse in Dutch maternity care. Lack of informed consent appears to be an important element of disrespect and abuse in maternity care in higher income settings, including in this study, and therefore requires special attention when conducting research in such settings. Caregivers' and birth companions' perspectives should also be investigated, to gain a more accurate understanding of the causes of this problem.

## Conclusion

This study gives insight into the content shared by women through the #genoeggezwegen campaign, that was started to break the silence on negative and traumatic experiences in Dutch maternity care. Women reported experiencing a lack of informed consent, not being taken seriously and not being listened to, lack of compassion and use of force; this left them feeling powerless. This may indicate that disrespect and abuse during labour and birth do happen in the Netherlands, although the current study gives no insight into prevalence. The findings of this study may contribute to better awareness of maternity care providers and the community on the possible existence of disrespect and abuse in Dutch maternity care, and encourage joint effort on improving care both individually and systemically/institutionally.

## Acknowledgments

We are grateful to the women who participated in the #genoeggezwegen campaign for sharing their stories.

## Author Contributions

**Conceptualization:** Marit S. G. van der Pijl, Martine H. Hollander, Tineke van der Linden, Rachel Verweij, Lianne Holten, Elselijn Kingma, Ank de Jonge, Corine J. M. Verhoeven.

**Data curation:** Marit S. G. van der Pijl, Tineke van der Linden, Rachel Verweij.

**Formal analysis:** Marit S. G. van der Pijl, Martine H. Hollander.

**Investigation:** Marit S. G. van der Pijl, Tineke van der Linden, Rachel Verweij, Lianne Holten, Corine J. M. Verhoeven.

**Methodology:** Marit S. G. van der Pijl, Martine H. Hollander, Tineke van der Linden, Rachel Verweij, Lianne Holten, Elselijn Kingma, Ank de Jonge, Corine J. M. Verhoeven.

**Project administration:** Marit S. G. van der Pijl, Corine J. M. Verhoeven.

**Supervision:** Martine H. Hollander, Lianne Holten, Ank de Jonge, Corine J. M. Verhoeven.

**Validation:** Elselijn Kingma.

**Writing – original draft:** Marit S. G. van der Pijl, Martine H. Hollander, Tineke van der Linden, Rachel Verweij, Lianne Holten, Elselijn Kingma, Ank de Jonge, Corine J. M. Verhoeven.

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
