## [Decision Letter · Decision Letter 0]

10 Feb 2020

PONE-D-19-33121

Left powerless: a social media content analysis of the Dutch #breakthesilence campaign on negative and traumatic experiences of labour and birth

PLOS ONE

Dear Mrs van der Pijl,

Thank you for submitting your manuscript to PLOS ONE. After careful consideration, we feel that it has merit but does not fully meet PLOS ONE’s publication criteria as it currently stands. Therefore, we invite you to submit a revised version of the manuscript that addresses the points raised during the review process.

We would appreciate receiving your revised manuscript by Mar 26 2020 11:59PM. To enhance the reproducibility of your results, we recommend that if applicable you deposit your laboratory protocols in protocols.io, where a protocol can be assigned its own identifier (DOI) such that it can be cited independently in the future. For instructions see: http://journals.plos.org/plosone/s/submission-guidelines#loc-laboratory-protocols

We look forward to receiving your revised manuscript.

Kind regards,

Florian Fischer

Academic Editor

PLOS ONE

Journal Requirements:

2. In your data availability statement, you write, "All #genoeggezwegen files analysed for the study are available and can be found at the Facebook page of Birth Movement NL (https://www.facebook.com/pg/GeboorteBeweging/photos/?tab=album&album_id=1149487385089171&ref=page_internal)." We note that the content on this Facebook page is at least partially restricted. Please deposit your underlying data to a dedicated data repository. A list of potential repositories, including ones with mechanisms for data access control is available here: https://journals.plos.org/plosone/s/data-availability#loc-recommended-repositories. Alternatively, if the data cannot be deposited, please discuss (1) the reasons for the restrictions and (2) a data-sharing mechanism such that qualified researchers could request the data. Note that it is not sufficient for a single author to be the sole individual responsible for ensuring data access.

3. We note that one or more authors are affiliated with Hechte Band or Stichting Geboortebeweging. In your competing interests statement, please discuss the relationship with this company and organization to the topic of the study. Note that PLOS ONE's competing interests' policy requires authors to discuss anything that could be perceived as a competing interest. For more information on what to declare, see https://journals.plos.org/plosone/s/competing-interests#loc-what-to-declare.

Reviewers' comments:

Reviewer's Responses to Questions

**Comments to the Author**

1. Is the manuscript technically sound, and do the data support the conclusions?

Reviewer #1: Partly

Reviewer #2: Partly

2. Has the statistical analysis been performed appropriately and rigorously? 

Reviewer #1: N/A

Reviewer #2: No

3. Have the authors made all data underlying the findings in their manuscript fully available?

Reviewer #1: Yes

Reviewer #2: No

4. Is the manuscript presented in an intelligible fashion and written in standard English?

Reviewer #1: Yes

Reviewer #2: Yes

5. Review Comments to the Author

Reviewer #1: Lines 86 and 585 note that women were asked to write their experiences "on a piece of paper". If this was the process, who collected and posted responses online? It may be that they were asked to post their experiences (along with the hashtag) on social media? If not, who asked them to write it on the paper? Where? Who was asked? Also, it's at this juncture that I'm wondering what platform(s) was/were suggested (i.e. facebook, twitter, instagram, etc.). Similarly, at line 123 it is noted that stories were collected from FB but also via email? This further confuses where/how stories were collected by the movement. The campaign process requires some clarity in order to inform data collection. Line 136 - I'm assuming that identifiable information in the women's stories as well as social media platform id's were also removed/deidentified? Also, at this point in the manuscript I'm a bit confused as to whether you're assessing images and text or just text. Beginning at line 466, the authors bring in another study regarding higher income settings - This is not well integrated and it seems out of place in the discussion. "Women reported a lack of informed consent, not being taken seriously and not being listened to, lack

630 of compassion and use of force; this left them feeling powerless. This indicates that disrespect and

631 abuse during labour and birth do happen in the Netherlands, although the current study gives no

632 insight into prevalence" - This study doesn't confirm disrespect and abuse, it only confirms that the women perceived their experiences as such. One of the limitations, as noted, as that we don't know the circumstances/timing/etc. of the experiences. Women's perceptions of their experiences are no less valuable but we have to be careful what we claim from the findings. This is prevalent throughout. The paper can be framed to describe women's perceptions of care rather than proof of abuse.

Reviewer #2: This is an interesting and informative piece of scholarship that uses social media content to contribute to understanding of traumatic labor and birth experiences. In many respects, the manuscript is outside my area of expertise, so I am unable to comment on some of the substance. My main qualification to review this manuscript is in terms of methods. Therefore, I am going to focus my review on that component of the research. While I do see some issues that are important and must be addressed, I ultimately think they are issues that can be sorted out.

I am concerned that the term “content analysis” is misused in the methods section. The term “content analysis” is typically reserved for quantitative approaches, as is described in the following works: .

Krippendorff, K. H. (2013). Content Analysis: An Introduction to Its Methodology (Third Edition). Sage Publications, Inc.

Neuendorf, K. A. (2017). The content analysis guidebook (Second edition). SAGE.

Riffe, D., Lacy, S., & Fico, F. G. (2005). Analyzing Media Messages: Using Quantitative Content Analysis in Research (2nd ed.). Lawrence Erlbaum Associates.

The qualitative process described here should be referred to as “textual analysis” or “qualitative content analysis” or by some other term to avoid confusion. I will note that it is referred to in these kinds of terms in other places in the manuscript, so this should frankly be an easy fix.

This brings me to my second – and perhaps more substantive – critique of the work. While it is billed as a qualitative analysis, extensive quantitative results are reported. It is is generally considered to be inappropriate to report figures from a qualitative textual analysis as though a quantitative process was conducted, because quantitative content analysis has certain requirements to ensure reliability and validity.

The use of MAXQDA Analytics Pro software is mentioned, but it is unclear what that software was used for. Was it used to automate the analysis process, or was it simply the platform on which coding was conducted? If human-coded quantitative content analysis was conducted, then intercoder reliability scores MUST be reported. If an algorithmic content analysis was conducted, it must be described in more detail.

I am unclear what this passage means on p. 7 (Line 155): “… a conventional content analysis, in which categories were derived from the data…” Generally speaking, in a conventional quantitative content analysis, coding categories are determined before data are collected. The authors may wish to briefly (one paragraph or so) reflect on the strengths and weaknesses of inductive vs. deductive processes in textual analysis.

Finally, I have a few other general comments:

-- The authors write: “Five major themes emerged from the stories.” How were these this determined? There needs to be more transparency about how these themes were decided upon.

-- While it is backed up by a citation, the statement “Currently, little use is made of social media content as data for health care research, despite its great potential due to the huge volume of health care experiences shared online” seems implausible. A closer look at the Greaves citation shows that it is an opinion essay rather than an empirical study, so I don’t think this statement is supported by the citation provided.

-- Similarly, the authors argue that “social media research” is “a relatively new and innovative field.” This statement is simply not true – the phenomenon of “social media” is at the center of vast quantities of research, including specifically in health care contexts. Furthermore, this research does not use any particularly innovative techniques (i.e. big data analysis). This does not mean that the research is bad! Quite the contrary, it appears to be good, useful research. My point is that the authors’ argument that this is groundbreaking or innovative research detracts and is not necessary.

-- This following passage doesn’t seem to make sense and should be cleaned up: “… they should not make us doubt that the #GG stories give good, and rare, first-person insight into how women were in fact affected and felt .”

6. PLOS authors have the option to publish the peer review history of their article (what does this mean?). If published, this will include your full peer review and any attached files.

Reviewer #1: No

Reviewer #2: No

---

## [Author Response · Author response to Decision Letter 0]

23 Mar 2020

A detailed response to the comments of the reviewers and editor can be found in the document 'Response to Reviewers'.

---

## [Decision Letter · Decision Letter 1]

23 Apr 2020

PONE-D-19-33121R1

Left powerless: a qualitative social media content analysis of the Dutch #breakthesilence campaign on negative and traumatic experiences of labour and birth

PLOS ONE

Dear Mrs van der Pijl,

Thank you for submitting your manuscript to PLOS ONE. The paper is almost suitable for publication, but we require to respond to one comment raised by or review (see mail below). Therefore, we invite you to submit a revised version of the manuscript that addresses the points raised during the review process.

We would appreciate receiving your revised manuscript by Jun 07 2020 11:59PM. To enhance the reproducibility of your results, we recommend that if applicable you deposit your laboratory protocols in protocols.io, where a protocol can be assigned its own identifier (DOI) such that it can be cited independently in the future. For instructions see: http://journals.plos.org/plosone/s/submission-guidelines#loc-laboratory-protocols

We look forward to receiving your revised manuscript.

Kind regards,

Florian Fischer

Academic Editor

PLOS ONE

Reviewers' comments:

Reviewer's Responses to Questions

**Comments to the Author**

1. If the authors have adequately addressed your comments raised in a previous round of review and you feel that this manuscript is now acceptable for publication, you may indicate that here to bypass the “Comments to the Author” section, enter your conflict of interest statement in the “Confidential to Editor” section, and submit your "Accept" recommendation.

Reviewer #2: All comments have been addressed

2. Is the manuscript technically sound, and do the data support the conclusions?

Reviewer #2: Yes

3. Has the statistical analysis been performed appropriately and rigorously? 

Reviewer #2: Yes

4. Have the authors made all data underlying the findings in their manuscript fully available?

Reviewer #2: Yes

5. Is the manuscript presented in an intelligible fashion and written in standard English?

Reviewer #2: Yes

6. Review Comments to the Author

Reviewer #2: As I noted in my earlier review, I am not a subject area expert on this topic, but I do have expertise in the methods used. I am impressed with the way the authors responded to my critiques of their work. The description methods is much more transparent now. Not only does it bolster confidence in the work reported here, but it now gives a good roadmap to people who might wish to do similar work in the future. This is a very well-written and compelling article, and I'm glad that I got the opportunity to review it.

The only aspect that gives me some pause is the passage: "In total 533 stories were collected, of which 95 were excluded." Why were these 95 excluded? Earlier you write that stories "shared by caregivers, partners or other witnesses of negative or traumatic experiences were excluded." Does that account for the 95? I might recommend adding just a few words about those exclusions to the passage quoted above, connecting it to that earlier explanation.

7. PLOS authors have the option to publish the peer review history of their article (what does this mean?). If published, this will include your full peer review and any attached files.

Reviewer #2: No

---

## [Author Response · Author response to Decision Letter 1]

24 Apr 2020

Dear Mr Fischer,

Hereby we resubmit our manuscript. Our detailed response can be found in the document 

'Response to Reviewers'.

Kind regards, on behalf of all authors,

Marit van der Pijl

---

## [Editor Report · Decision Letter 2]

29 Apr 2020

Left powerless: a qualitative social media content analysis of the Dutch #breakthesilence campaign on negative and traumatic experiences of labour and birth

PONE-D-19-33121R2

Dear Dr. van der Pijl,

We are pleased to inform you that your manuscript has been judged scientifically suitable for publication and will be formally accepted for publication once it complies with all outstanding technical requirements.

With kind regards,

Florian Fischer

Academic Editor

PLOS ONE
---

## [Editor Report · Acceptance letter]

1 May 2020

PONE-D-19-33121R2 

Left powerless: a qualitative social media content analysis of the Dutch #breakthesilence campaign on negative and traumatic experiences of labour and birth 

Dear Dr. van der Pijl:

I am pleased to inform you that your manuscript has been deemed suitable for publication in PLOS ONE. Congratulations! Your manuscript is now with our production department. 

With kind regards,

on behalf of

Dr. Florian Fischer 

Academic Editor

PLOS ONE